# Epidemiology of hepatitis B virus and/or hepatitis C virus infections among people living with human immunodeficiency virus in Africa: A systematic review and meta-analysis

Raoul Kenfack-Momo[1], Sebastien Kenmoe[2,3] *, Guy Roussel Takuissu[4], Jean Thierry Ebogo-Belobo[4], Cyprien Kengne-Ndé[5], Donatien Serge Mbaga[6], Serges Tchatchouang[7], Martin Gael Oyono[4], Josiane Kenfack-Zanguim[1], Robertine Lontuo Fogang[8], Chris Andre Mbongue Mikangue[6], Elisabeth Zeuko'o Menkem[9], Juliette Laure Ndzie Ondigui[6], Ginette Irma Kame-Ngasse[4], Jeannette Nina Magoudjou-Pekam[1], Jean Bosco Taya-Fokou[6], Arnol Bowo-Ngandji[6], Seraphine Nkie Esemu[2], Diane Kamdem Thiomo[1], Paul Moundipa Fewou[1], Lucy Ndip[2], Richard Njouom[3]

1 Department of Biochemistry, The University of Yaounde I, Yaounde, Cameroon, 2 Department of Microbiology and Parasitology, University of Buea, Buea, Cameroon, 3 Virology Department, Centre Pasteur of Cameroon, Yaoundé, Cameroon, 4 Centre for Research on Health and Priority Pathologies, Institute of Medical Research and Medicinal Plants Studies, Yaounde, Cameroon, 5 Epidemiological Surveillance, Evaluation and Research Unit, National AIDS Control Committee, Douala, Cameroon, 6 Department of Microbiology, The University of Yaounde I, Yaounde, Cameroon, 7 Scientific Direction, Centre Pasteur of Cameroon, Yaounde, Cameroon, 8 Department of Animal Biology, University of Dschang, Dschang, Cameroon, 9 Department of Biomedical Sciences, University of Buea, Buea, Cameroon

* sebastien.kenmoe@ubuea.cm

## Abstract

### Introduction

Due to their common routes of transmission, human immunodeficiency virus (HIV) coinfection with hepatitis B virus (HBV) and/or hepatitis C virus (HCV) has become a major public health problem worldwide, particularly in Africa, where these viruses are endemic. Few systematic reviews report the epidemiological data of HBV and/or HCV coinfection with HIV in Africa, and none provided data on the case fatality rate (CFR) associated with this coinfection. This study was conducted to investigate the prevalence and case fatality rate of HBV and/or HCV infections among people living with human immunodeficiency virus (PLHIV) in Africa.

### Methods

We conducted a systematic review of published articles in PubMed, Web of Science, African Journal Online, and African Index Medicus up to January 2022. Manual searches of references from retrieved articles and grey literature were also performed. The meta-analysis was performed using a random-effects model. Sources of heterogeneity were investigated using subgroup analysis, while funnel plots and Egger tests were performed to assess publication bias.

**Data Availability Statement:** All relevant data are within the manuscript and its Supporting Information files.

**Funding:** This project is part of the EDCTP2 programme supported by the European Union under grant agreement TMA2019PF-2705. "The funders had no role in study design, data collection and analysis, decision to publish, or preparation of the manuscript."

**Competing interests:** The authors have declared that no competing interests exist.

## Results

Of the 4388 articles retrieved from the databases, 314 studies met all the inclusion criteria. The overall HBV case fatality rate estimate was 4.4% (95% CI; 0.7–10.3). The overall sero-prevalences of HBV infection, HCV infection, and HBV/HCV coinfection in PLHIV were 10.5% [95% CI = 9.6–11.3], 5.4% [95% CI = 4.6–6.2], and 0.7% [95% CI = 0.3–1.0], respectively. The pooled seroprevalences of current HBsAg, current HBeAg, and acute HBV infection among PLHIV were 10.7% [95% CI = 9.8–11.6], 7.0% [95% CI = 4.7–9.7], and 3.6% [95% CI = 0.0–11.0], respectively. Based on HBV-DNA and HCV-RNA detection, the sero-prevalences of HBV and HCV infection in PLHIV were 17.1% [95% CI = 11.5–23.7] and 2.5% [95% CI = 0.9–4.6], respectively. Subgroup analysis showed substantial heterogeneity.

## Conclusions

In Africa, the prevalence of hepatotropic viruses, particularly HBV and HCV, is high in PLHIV, which increases the case fatality rate. African public health programs should empha-size the need to apply and comply with WHO guidelines on viral hepatitis screening and treatment in HIV-coinfected patients.

## Review registration

PROSPERO, CRD42021237795.

## Introduction

Improved care and access to antiretroviral treatment has led to a significant increase in the life expectancy of people living with HIV (PLHIV) [1]. Currently, there are approximately 40 million PLHIV worldwide. The United Nations Programme on HIV/AIDS (UNAIDS) reported in 2020 that more than one person dies every minute from AIDS-related illnesses [2]. HIV, hepatitis B virus (HBV) and hepatitis C virus (HCV) have several common routes of transmis-sion, including parenteral, sexual and vertical transmission [3]. PLHIV are frequently coin-fected with HBV and/or HCV [4, 5]. Liver diseases are one of the main causes of death in PLHIV [6, 7]. There are more than 2 million people infected with HBV HCV among PLHIV globally [4, 5, 8]. The high morbidity and mortality of HBV and/or HCV coinfection in PLHIV are explained by bidirectional effects. PLHIV coinfected with HBV and/or HCV show rapid progression to AIDS [9] and are also associated with increased hepatic toxicity of antire-trovirals [10–12]. Conversely, the alteration of the immune response in PLHIV leads to a decrease in HBV and/or HCV viral clearance, reactivation, and an increase in viral replication in coinfected patients. This leads to an increase in liver enzyme levels (aspartate aminotrans-ferase, alanine aminotransferase and alkaline phosphatase), a faster evolution towards chronic-ity, complications of liver diseases (cirrhosis, hepatic decompensation, hepatocellular carcinoma), and a rising mortality [13–15]. PLHIV coinfected with HBV and/or HCV have a higher risk of transmission of infection. In addition, the drug interaction between treatments overlapping HIV and HBV infections is subject to the selection of strains resistant to HBV treatment [16, 17]. HBV and/or HCV coinfection in PLHIV receives considerable public health attention due to its high burden and its negative impact on the survival of affected patients. Several evidence-based syntheses have already been conducted to characterize the

categories of populations most at risk to inform priorities in control and treatment [4, 5]. Coinfection rates vary significantly, and parameters such as geographical region and population categories represent the main risk factors. Agreeingly, the data syntheses show that the most affected groups are injecting drug users (IDU) and men who have sex with men (MSM). Africa is the most endemic area for HBV and/or HCV coinfection in PLHIV, and more than 2-thirds of the 37.7 million PLHIV reside in sub-Saharan Africa [2]. Despite the availability of these data syntheses that show the high prevalence of HBV and/or HCV in PLHIV, it remains difficult to apply the WHO guidelines on the screening and management of HBV and/or HCV coinfections in PLHIV in low-resource settings in Africa [18]. To date, there is no summary of evidence reporting the case fatality rate (CFR) in PLHIV coinfected with HBV and/or HCV. This study was initiated to update HBV and/or HCV prevalence and report the case fatality rate (CFR) of HBV and/or HCV coinfection in PLHIV in Africa.

## Methods

### Study design and search strategy

The Preferred Reporting Items for Systematic Reviews and Meta-Analysis (PRISMA) standard was used in the design of this review (S1 Table) [19]. We declared this study to the International Prospective Register of Systematic Reviews (PROSPERO) under the registration number CRD42021237795. We applied our search strategy (S2 Table) in 4 databases (PubMed, Web of Science, African Journal Online and African Index Medicus) to identify all relevant articles containing data on the prevalence and/or CFR of HBV and/or HCV in PLHIV. We applied no time restriction. The first search was conducted in February 2021, and we conducted an updated search in January 2022. We manually searched for the references of relevant articles and previous systematic reviews to identify any additional articles missed by the online search [20–22].

### Selection of studies

The articles found were exported to Endnotes version X9 software, from which duplicates were eliminated. We (JETB and SK) independently selected articles based on their titles and/or abstracts on the Rayyan review platform. We (14 review authors) obtained the full texts and screened potentially eligible articles from this review for final inclusion. We resolved differences through discussion and consensus.

### Inclusion and exclusion criteria

We selected articles that met the following criteria: 1) conducted only on the African continent and limited to human studies; 2) reporting the prevalence of HBV and/or HCV among PLHIV, without restriction on age, sex, study design, sampling approach, year of publication, type of HIV/HCV/HBV detection assays, and sample type. We excluded publications with less than 10 PLHIV or that did not provide data on HBV and/or HCV prevalence or CFR among PLHIV. We also excluded studies with unavailable abstracts or full texts and duplicates.

### Data extraction

Google forms were used to extract data on the first author's name, year of publication, study period, study design, sampling approach, number of sites (monocenter, multicenter or nationally representative), timing of participant recruitment (prospective, retrospective or retroprospective), country, United Nations Statistics Division (UNSD) region, WHO region, UNAIDS region, country income level, setting (community or hospital and urban and rural), age group, antiretroviral therapy (ART) status, PLHIV category, HBV and/or HCV diagnostic assay,

diagnostic target, sample size, number of PLHIV positive for HBV and/or HCV, and the risk of bias assessed according to the Hoy *et al* tool (S3 Table) [23].

## Data synthesis and analysis

We estimated the prevalence and 95% confidence intervals (95% CI) using the total size and number of HBV and/or HCV positives via a random-effect model and using R software version 4.1.0 [24, 25]. We used the same meta-analysis to estimate the CFR (95% CI) from the number of deaths among HBV- and/or HCV-positive PLHIV. We assessed heterogeneity by the Cochrane Q statistic χ2 test and used H and I2 values for quantification [26]. H is a measure of the extent of heterogeneity, and a value of H > 1 indicates potential heterogeneity of effects. I2 describes the proportion of total variation between the included studies, and a value > 50% indicates the presence of heterogeneity [26, 27]. We investigated sources of heterogeneity using subgroup analysis according to study design, sampling method, time of PLHIV recruitment, country, United Nations Statistics Division (UNSD) region, level country income, setting, age group, ART status, and PLHIV category. We used a funnel plot and Egger's test to assess publication bias [28].

## Results

### Results of the study search

The flowchart of the article selection process is shown in **Fig 1**. A total of 6375 articles were found in the databases, and 2223 duplicates were deleted. An additional 237 articles were obtained from other sources. The titles and abstracts assessment of the remaining 4388 articles resulted in the exclusion of 3793. Then, after a full-text review, another 279 articles were excluded for multiple reasons (**S4 Table**). For this systematic review, a total of 314 articles (568 prevalence data) met the inclusion criteria (**S1 Text**).

### Characteristics of included studies

The characteristics of the included studies are depicted in S5 and S6 Tables. Of the 568 prevalence data included in this review, cross-sectional (81.2%) and prospective (79.9%) studies were the main study designs, while nonprobabilistic (85.2%) and consecutive (79.6%) sampling were the most commonly used methods. All the articles were published between 1990 and 2022 in 34 countries in Africa (5 UNSD Regions). Most of the studies were from West Africa (44.0%), followed by Eastern Africa (30.5%), Southern Africa (10.0%), Central Africa (9.7%) and Northern Africa (2.3%). According to the countries, most studies came from Nigeria (21.3%) and South Africa (10.0%), while Benin (0.2%), Egypt (0.2%) and Burundi (0.2%) had the smallest numbers of studies. The participants were recruited between 1987 and 2022 and consisted mainly of adults (51.1%) and the general population (75.0%), of whom most were recruited in hospitals (95.8%). HBV and/or HCV diagnosis was mainly made by direct ELISA (20.8%) and rapid diagnostic tests (19.4%). The majority of studies indicated a low risk of bias (56.5%) (**S7 Table**).

### Case fatality rate estimate of hepatitis B and/or C virus infections in people living with HIV in Africa

The number of deaths due to HBV infection in PLHIV was reported in 05 studies from 4 countries (Kenya, Ivory Coast, Nigeria and Gambia) (Figs 2 and 3). Out of 2226 participants recruited, 68 deaths were recorded, giving an overall HBV fatality rate of 4.4% (95% CI; 0.7–10.3). This HBV-CFR was 0.0% (95% CI; 0.0–15.0) in Eastern Africa and 5.5% (95% CI; 1.2–

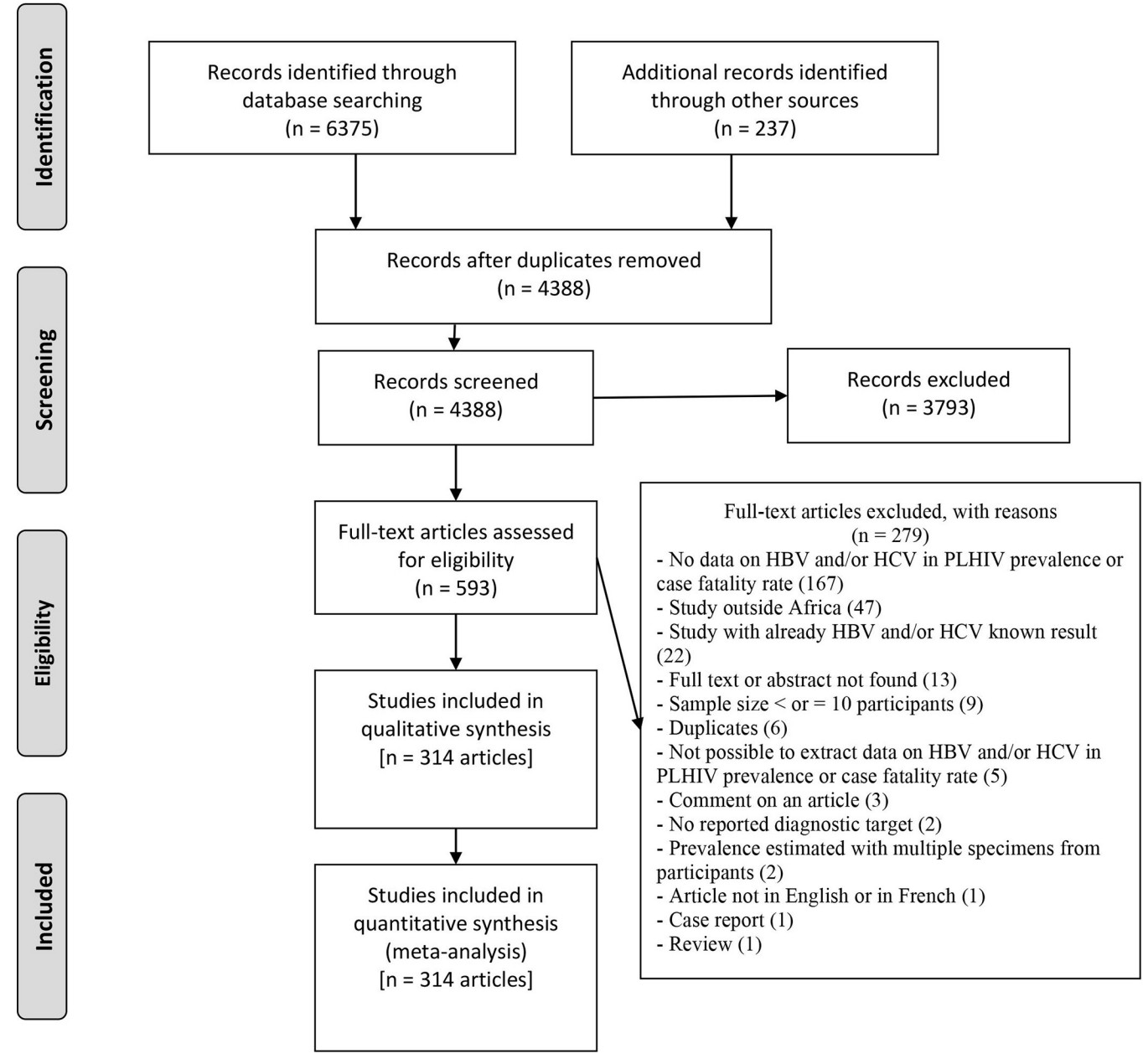

**Fig 1. Flowchart for retrieving and processing article selection.**

12.1) in West Africa. A statistically significant heterogeneity was observed in the overall estimate of the HBV CFR estimate ($I^2 = 83.4\%$ [62.4%; 92.7%], p < 0.0001). The review of Egger's regression test results (Table 1) and the funnel plot (S1 Fig) suggested no publication bias (P = 0.283). We obtained no data on the CFR of HCV in PLHIV.

## Pooled prevalence estimate of hepatitis B virus infection in PLHIV in Africa

A total sample size of 324305 participants from 32 countries was considered for HBV prevalence estimates in PLHIV in Africa (**Figs 2 and 4, S2 Fig**). The overall prevalence of HBV infection in PLHIV was 10.5% [95% CI = 9.6–11.3], with statistically significant heterogeneity

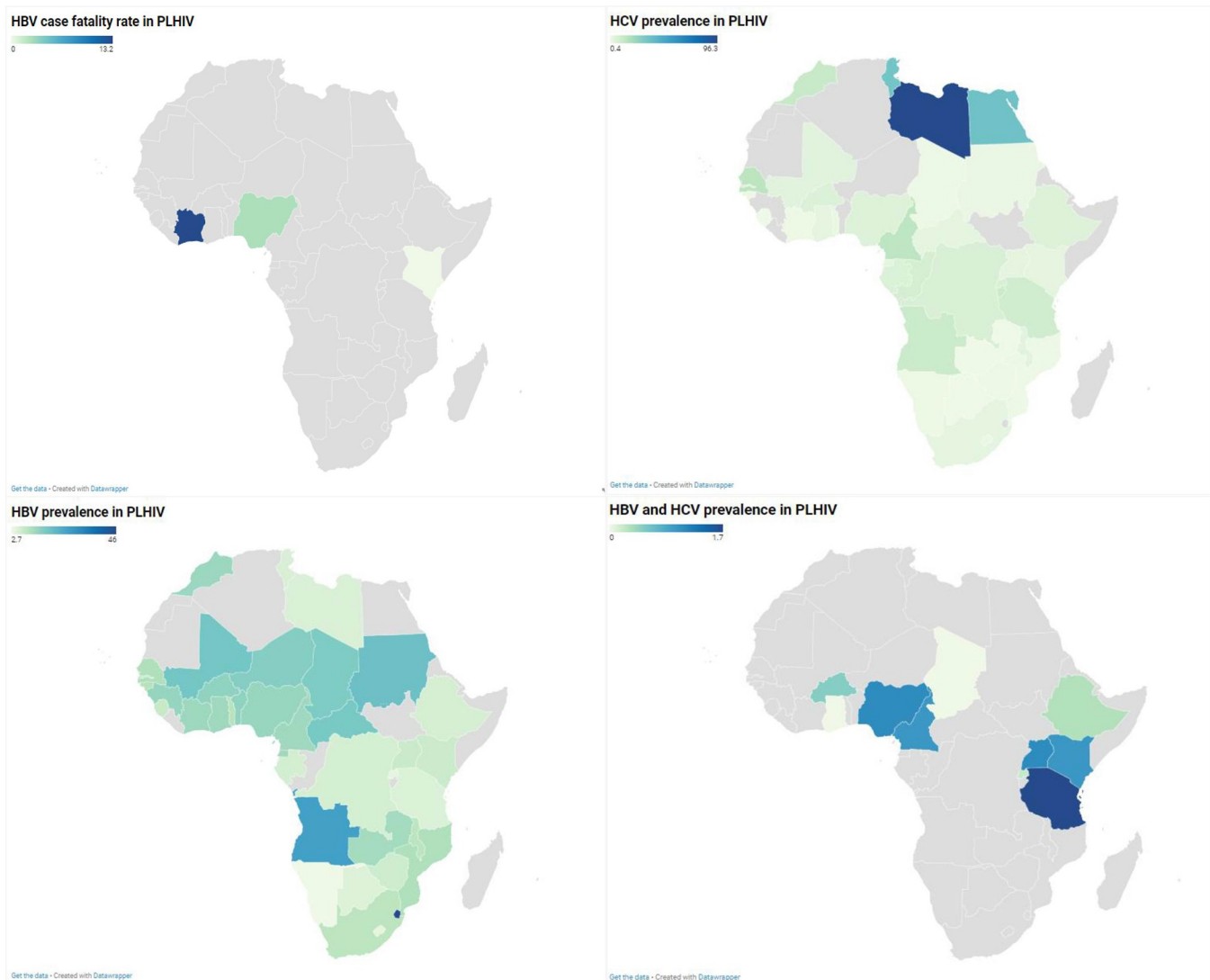

**Fig 2. HBV case fatality rate and prevalence estimates of HBV and/or HCV infections in PLHIV in Africa.**

($I^2$ = 98.0% [97.9%-98.1%], p < 0.001) (**Table 1**). Egger's test and funnel plot (**S3 Fig**) showed statistically significant publication bias (P < 0.001). The prevalence rates of current HBV infection (HBsAg), current HBV infectivity (HBeAg), and acute HBV infection (HBsAg + IgM anti−HBc) were 10.7% [95% CI = 9.8–11.6], 7.0% [95% CI = 4.7–9.7], and 3.6% [95% CI = 0.0–11.0], respectively. Twelve studies reported the HBV infection prevalence based on the detection of HBV DNA: 17.1% [95% CI = 11.4–23.7].

## Pooled prevalence estimate of hepatitis C virus infection in PLHIV in Africa

A total sample size of 255143 participants from 27 countries was considered for HCV prevalence estimates in PLHIV in Africa (**Figs 2 and 5, S4 Fig**). The overall prevalence of HCV infection in PLHIV was 5.4% [95% CI = 4.6–6.2], with statistically significant heterogeneity ($I^2$ = 98.0% [97.9%-98.2%], p < 0.01) (**Table 1**). Egger's test and funnel plot (**S5 Fig**) showed no

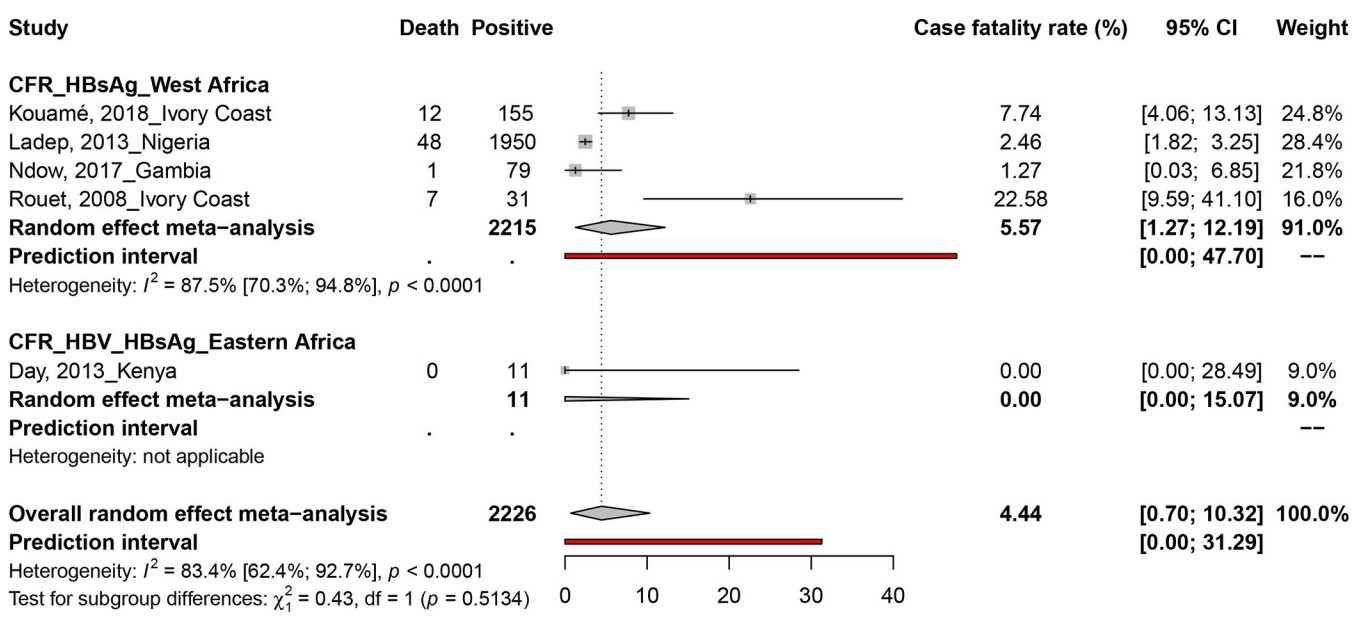

**Fig 3. Case fatality rate estimate of hepatitis B virus infections in people living with HIV in Africa.**

**Table 1. Summary of meta-analysis results for epidemiology of hepatitis B and C in people living with HIV in Africa.**

|  | Prevalence. % (95%CI) | 95% Prediction interval | N Studies | N Participants | ¶H (95%CI) | §I² (95%CI) | P heterogeneity |
|---|---|---|---|---|---|---|---|
| **HBV case fatality rate in PLHIV** |  |  |  |  |  |  |  |
| Overall | 4.4 [0.7–10.3] | [0–31.3] | 5 | 2226 | 2.5 [1.6–3.7] | 83.4 [62.4–92.7] | <0.001 |
| Low risk of bias | 5.9 [0.6–14.7] | [0–58.1] | 4 | 2147 | 2.8 [1.8–4.3] | 87.3 [69.7–94.7] | <0.001 |
| **HBV prevalence in PLHIV** |  |  |  |  |  |  |  |
| Overall | 10.5 [9.6–11.3] | [0.8–28.3] | 313 | 324305 | 7 [6.8–7.2] | 98 [97.9–98.1] | <0.001 |
| Cross-sectional | 10.5 [9.5–11.5] | [0.7–28.6] | 251 | 251909 | 6.7 [6.5–6.9] | 97.7 [97.6–97.9] | <0.001 |
| Low risk of bias | 10.5 [9.4–11.6] | [0.8–28.5] | 179 | 284626 | 8.5 [8.3–8.8] | 98.6 [98.5–98.7] | <0.001 |
| **HCV prevalence in PLHIV** |  |  |  |  |  |  |  |
| Overall | 5.4 [4.6–6.2] | [0–20.2] | 210 | 255143 | 7.1 [6.9–7.4] | 98 [97.9–98.2] | <0.001 |
| Cross-sectional | 5.3 [4.5–6.2] | [0–21.1] | 180 | 212692 | 7 [6.8–7.2] | 98 [97.8–98.1] | <0.001 |
| Low risk of bias | 4.9 [4.1–5.8] | [0–16.7] | 114 | 224779 | 7.4 [7.1–7.7] | 98.2 [98–98.3] | <0.001 |
| **HBV and HCV prevalence in PLHIV** |  |  |  |  |  |  |  |
| Overall | 0.7 [0.3–1] | [0–2.9] | 26 | 136528 | 3.3 [2.9–3.8] | 91 [88–93.2] | <0.001 |
| Cross-sectional | 0.8 [0.3–1.3] | [0–4.1] | 19 | 123479 | 3.4 [2.9–4] | 91.3 [87.9–93.8] | <0.001 |
| Low risk of bias | 0.4 [0.2–0.6] | [0–1.5] | 17 | 133637 | 2.7 [2.2–3.3] | 85.8 [78.8–90.6] | <0.001 |

CI: confidence interval; N: Number; 95% CI: 95% Confidence Interval; NA: not applicable. ¶H is a measure of the extent of heterogeneity; a value of H = 1 indicates homogeneity of effects, and a value of H >1 indicates a potential heterogeneity of effects. §: I2 describes the proportion of total variation in study estimates that is due to heterogeneity; a value > 50% indicates the presence of heterogeneity.

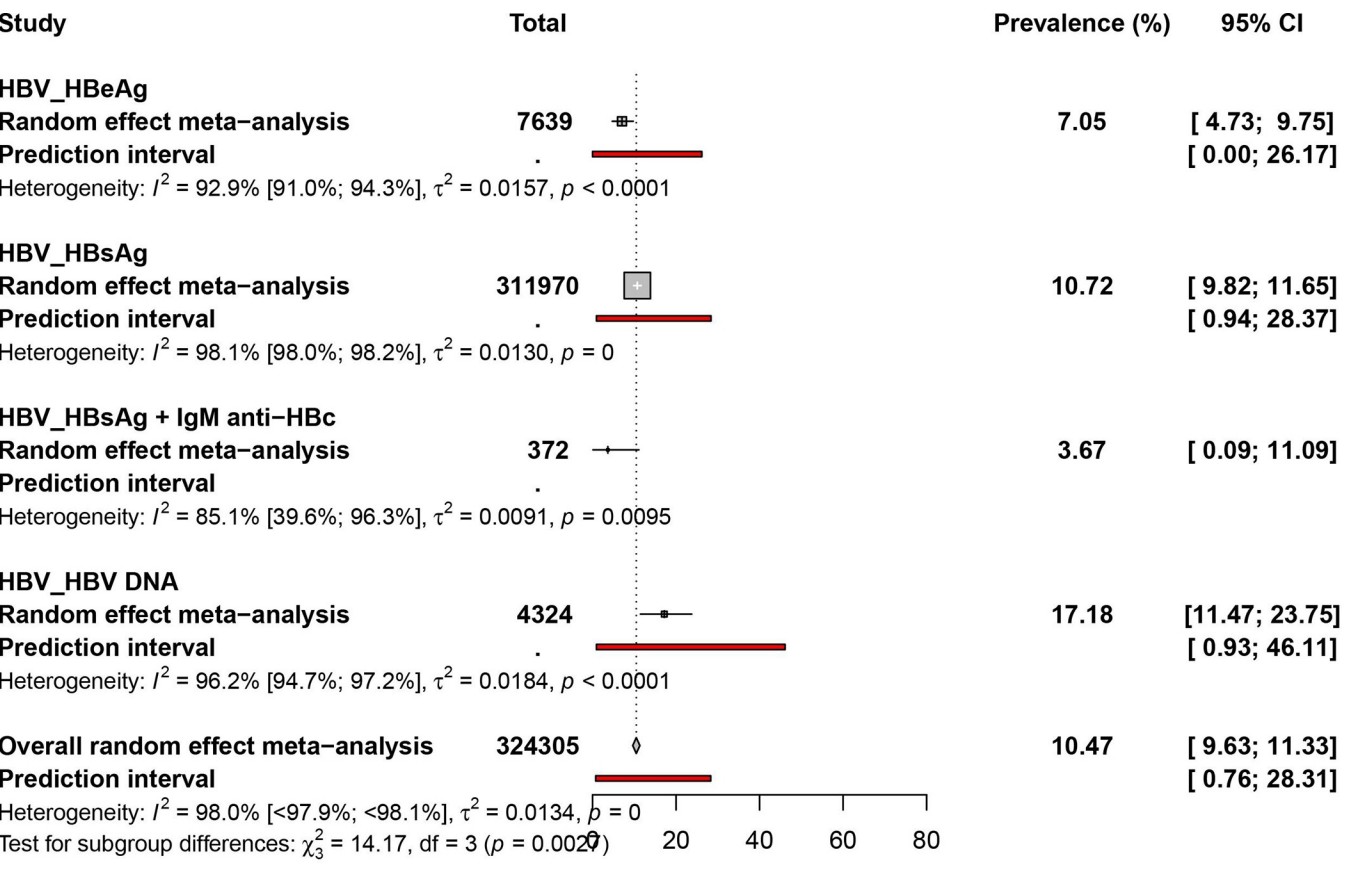

**Fig 4. Prevalence estimate of hepatitis B virus infection in PLHIV in Africa.**

publication bias (P = 0.642). One hundred and sixteen studies reported the anti-HCV seroprevalence as 5.6% [95% CI = 4.8–6.5], and 13 studies reported the prevalence from the detection of HCV RNA as 2.5% [95% CI = 0.9–4.6].

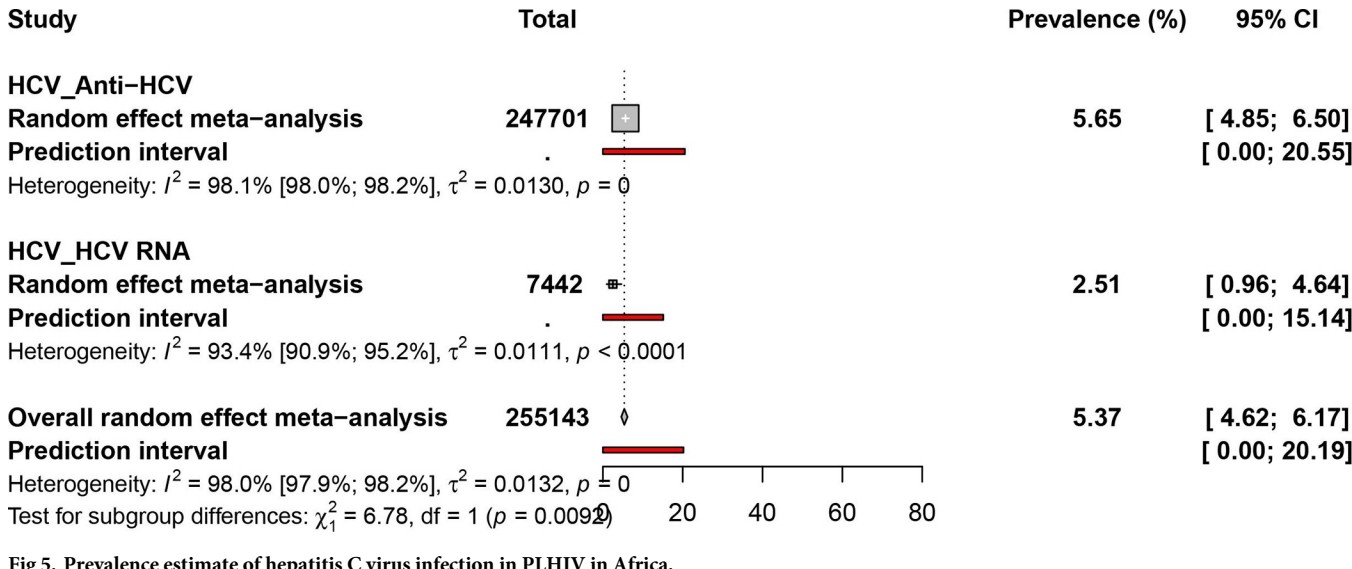

**Fig 5. Prevalence estimate of hepatitis C virus infection in PLHIV in Africa.**

## Pooled prevalence estimate of hepatitis B and C virus coinfection in PLHIV in Africa

A total sample size of 136528 participants from 10 countries were considered for HBV and HCV coinfection in Africa (Figs 2 and 6). The overall prevalence of HBV/HCV coinfection in PLHIV was 0.7% [95% CI = 0.3–1.0], with statistically significant heterogeneity ($I^2$ = 91.0% [88.0%-93.2%], p < 0.01) (Table 1). Egger's test and funnel plot (S6 Fig) showed significant publication bias (P = 0.004).

## Sensitivity analyses

Although about half of the studies had a moderate risk of bias, sensitivity analyzes showed no difference between the overall results and the results including only the studies with a low risk

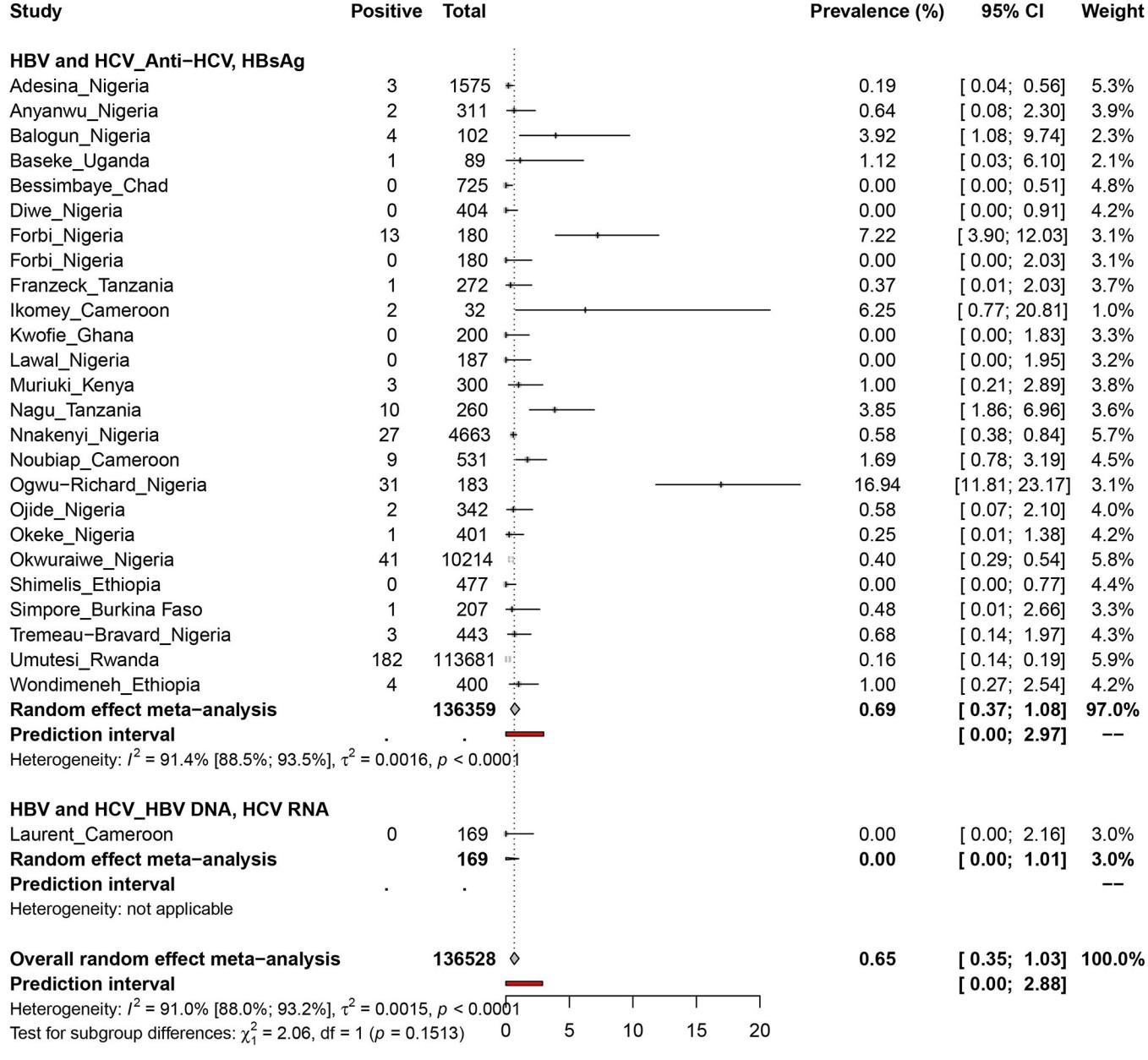

**Fig 6. Prevalence estimate of hepatitis B and C virus coinfection in PLHIV in Africa.**

of bias (Table 1). Overall results were also similar to results including only cross-sectional studies.

## Subgroup analyses

The results of the subgroup analysis performed in this study are shown in S8 Table and Fig 2. The HBV CFR was significantly higher in retrospective studies (22.6%; 95% CI = 9.3–39.2; p = 0,001) and in children (22.6%; 95% CI = 9.3–39.2; p = 0,002).

Subgroup analysis showed that HBV prevalence in PLHIV was significantly higher in case controls (13.3%; 95% CI = 9.4–17.8; p = 0.017), in studies with nonprobabilistic sampling (11%; 95% CI = 10.1–12; p<0.001), in Burkina Faso (14.7%; 95% CI = 12.1–17.4), Nigeria (13.3%; 95% CI = 11.6–15), Ivory Coast (13.1%; 95% CI = 10.2–16.3), Cameroon (12.6%; 95% CI = 9.9–15.7), Ghana (12.5%; 95% CI = 9.6–15.8), and Zambia (12.1%; 95% CI = 10.3–14.1; p<0.001), in West and Central UNAIDS region (12.5%; 95% CI = 11.5–13.5; p<0.001), in West Africa (12.8%; 95% CI = 11.7–14), Northern Africa (11.5%; 95% CI = 5–20.1), and Central Africa UNSD regions (11.2%; 95% CI = 8.9–13.6; p<0.001), lower-middle-income economies (11.5%; 95% CI = 10.5–12.6; p<0.001), adults (11.3%; 95% CI = 10.2–12.5; p<0.001), ART naïve (12.2%; 95% CI = 9.5–15.3; p<0.022), blood donors (15%; 95% CI = 11.9–18.4; p = 0.001), in studies that used classical PCR (19.7%; 95% CI = 8.5–33.9),immunoassay kit (14.7%; 95% CI = 11–18.9; p<0.001)and HBV DNA (17.2%; 95% CI = 11.5–23.8; p = 0.003).

Subgroup analysis also showed that HCV prevalence in PLHIV was significantly higher in studies with probabilistic recruitment (10.9%; 95% CI = 4.7–19; p = 0.042), Cameroon (13.9%; 95% CI = 7.1–22.4), Tanzania (9.2%; 95% CI = 2.9–18.3; p<0.001), North Africa and the Middle East UNAIDS region (30%; 95% CI = 8.2–58.1; p = 0.006), the Eastern Mediterranean WHO region (30%; 95% CI = 8.2–58.1; p = 0.016), the Northern Africa UNSD region (30%; 95% CI = 8.2–58.1; p<0.001), lower-middle income economies (6.4%; 95% CI = 5.2–7.7; p = 0.008), community-based studies (19.2%; 95% CI = 9.4–31.3; p = 0.001), urban settings (55%; 95% CI = 39.3–70.2; p<0.001), and IDU (58%; 95% CI Additionally, studies that used classical RT–PCR (33.3%; 95% CI = 0–99.6), indirect ELISA (7%; 95% CI = 5.9–8.2; p<0.001), and anti-HCV (5.6%; 95% CI = 4.8–6.5; p = 0.010) reported a higher prevalence of HCV in PLHIV.

These analyses also revealed HBV and HCV coinfection prevalence in PLHIV to be significantly higher in Tanzania (1.7%; 95% CI = 0–6.6; p<0.001), lower-middle income economies (1.1%; 95% CI = 0.5–1.8; p = 0.002), adults (1.7%; 95% CI = 0.7–3.1; p = 0.019), community-based studies (2.5%; 95% CI = 2.3–2.8; p<0.001), and studies that used immunochromatographic tests (3.9%; 95% CI = 0–14.4; p = 0.001).

## Discussion

To the best of our knowledge, this review is the first to address the synthesis of data on CFR due to HBV and/or HCV coinfection in PLHIV in Africa. Our systematic review is also an update of data on HBV and/or HCV coinfection in PLHIV in Africa. The study included 314 articles published between 1990 and 2022 in 34 African countries. Only 5 studies revealed data on CFR due to HBV coinfection in PLHIV in Africa, with a frequency of 4.4%. The overall prevalence of HBV among PLHIV in Africa was 10.5% and was higher in ARV treatment-naïve patients, in West Africa, Northern Africa, and Central Africa UNSD regions, in adults, and in blood donors. The overall prevalence of HCV among PLHIV in Africa was 5.4% and was higher in the Northern Africa UNSD region and in IDU. The overall prevalence of HBV and HCV coinfection among PLHIV in Africa was 0.7%.

A relatively high CFR of 4.4% in HIV/HBV coinfected subjects was recorded in this systematic review. Multiple studies have previously demonstrated higher morbidity and mortality in HIV/HBV coinfected subjects compared to HBV or HIV monoinfected subjects. This is explained by harmful synergistic interactions between the two viruses leading to a rapid progression towards AIDS-defining illnesses and increased hepatotoxicity of ARVs on one hand and an earlier development of hepatocellular carcinoma and the emergence of strains resistant to HBV treatment on the other hand [9–17, 29–32]. Longitudinal studies that would contribute to identifying the factors associated with the high morbidity and/or mortality of HBV and/or HCV coinfections in PLHIV in Africa would also be of great added value for the fight against these diseases in Africa, this context characterized particularly by its limited resources [33]. The prevalence of approximately 10.5% determined in the present work is similar to regional data obtained in a recent global review (10%) and lower than a previous study conducted in sub-Saharan Africa (15%) [4, 21]. The 10.5% prevalence of HIV/HBV coinfection determined in our review is higher than WHO estimates for the general African population, which is known to range between 5–8%. This confirms the higher risk of HBV contraction in PLHIV [3]. This high risk of HBV transmission in PLHIV is mainly explained by the common transmission routes of these 2 viruses. Global data showed high sexual transmission risks among sex workers and MSM and through injection among IDU [4]. The majority of studies conducted in Africa focus on nonspecific populations and to a lesser extent on blood donors and pregnant women. HBV is mainly transmitted in areas of high endemicity, such as Africa, in children under 5 years old through vertical and horizontal transmission. The risks of progression to chronicity are also very high in children compared to adults. HIV, on the other hand, is mainly acquired in Africa through sexual intercourse in adulthood. We can therefore hypothesize that in Africa, most of the HBV/HIV coinfected subjects come from an infection of HBV monoinfected subjects acquired in childhood who will become HIV coinfected in adulthood through the sexual route. Additional data on the routes of transmission of HIV, HBV and HCV in Africa are awaited. Although this specific transmission pattern in Africa of HBV and HIV in childhood and adulthood respectively is true for most cases, it is evident that many other reasons not fully described and/or clear exist and warrant more attention. Specific studies on subjects at high risk of HBV and/or HCV coinfections (sex workers, MSM, and IDU) among PLHIV are needed. The sustainability and/or strengthening of the protection of newborns through vaccination programs remains crucial for reducing mortality from HBV and/or HCV coinfections in PLHIV. We found a higher HBV prevalence in ART-naïve PLHIV. This result suggests that to prevent HBV infections, PLHIV should use ARV early. The pooled HCV prevalence found in this study (5.4%) is higher than that obtained previously in sub-Saharan Africa (2.98%) [22]. This HCV prevalence is very close to that of the general African population estimated at 5.3%, which shows that the prevalence of HCV increases with HIV coinfection [34, 35]. Furthermore, it is lower than the overall HCV prevalence reported in Eastern Europe and Central Asia (27.0%) [5]. These contrasting results may be due to the differences in the HCV transmission routes. Indeed, as reported in this review, IDU showed higher HCV prevalence (58%), while Eastern Europe and Central Asia are the areas with the highest number of PLHIV who inject drugs [36]. Indeed, previous studies have already shown that HIV-HCV co-infection prevalence is very high in IDU [37]. Only 04 studies from a single country (Kenya) reported the prevalence of HIV-HCV co-infection in IDU in this study. Data on IDU remains really scarce in Africa. Indeed, due to stigma, high crime and incarceration rates, mental health disorders, and unstable housing, IDU are understudied in sub-Saharan Africa (SSA) [38–40]. Yet, several authors report that number of IDU is on the rise in countries across SSA, especially in large urban settings [41, 42]. In Africa, it would be important to implement programs for the management of HCV/HIV coinfected IDU. Although the main

transmission route of HCV in countries of low and middle income remains poorly understood, several studies show that this transmission is associated with some medical procedures (blood transfusions, dialysis, unsterile injections) and cultural practices (barbering, tattooing, and body piercing) [43]. Given that in this review, the studies that reported the prevalence of HIV/HCV coinfection were mainly conducted in the general population, we recommend that future studies focus on at-risk populations such as hemodialysis patients, MSM, children with sickle cell disease, hemophilia and those with beta-thalassemia. According to the UNSD region, the highest HCV/HIV coinfection was found in Northern Africa, which agrees with previous reports [20]. Indeed, this region bears the greatest burden of HCV infection worldwide, with a peak of 14.7% found in Egypt [44]. Thus, Northern Africa is an area of priority when strengthening the effective implementation of hepatitis C virus screening and treatment for PLHIV. The prevalence of HCV reported in this review was mainly focused on anti-HCV detection. However, the diagnosis of HCV infection using anti-HCV screening tests has been demonstrated to show high rates of false-positive results, which tends to increase prevalence. These false reactions have been assigned to cross-reactivity with other types of antibodies produced by polyclonal B cells from exposure to some endemic infections, such as malaria, sleeping sickness and schistosomiasis [45]. Considering the low number of studies that reported data on RT–PCR-confirmed HCV infections in this report, we suggest that future studies should confirm their HCV antibody test results with molecular or antigen-based assays.

Although this systematic review is the first to report HBV-CFR in PLHIV and one of the few to synthesize data on HIV/HBV/HCV triple infection in Africa, there are some limitations. Our HBV-CFR and HBV and/or HCV prevalence estimates in PLHIV showed substantial heterogeneity between studies and publication bias. In addition, we have restricted our inclusion criteria only to English or French. However, we excluded only one study for this reason.

In addition to the fact that HIV is endemic in African regions, we can conclude that many PLHIV are HBV and/or HCV coinfected, suggesting a substantial burden of coinfection in this resource-poor setting. This coinfection with hepatotropic viruses significantly impacted the survival of PLHIV and requires establishing measures to reduce the risks of liver-related complications such as cirrhosis and hepatocellular carcinomas.

## Supporting information

**S1 Checklist.**
(PDF)

**S1 Table. Preferred reporting items for systematic reviews and meta-analyses checklist.**
(PDF)

**S2 Table. Search strategy in PubMed.**
(PDF)

**S3 Table. Items for risk of bias assessment.**
(PDF)

**S4 Table. Main reasons for exclusion of eligible studies.**
(PDF)

**S5 Table. Characteristics of included studies.**
(PDF)

**S6 Table. Individual characteristics of included studies.**
(PDF)

**S7 Table. Risk of bias assessment.**
(PDF)

**S8 Table. Subgroup analyses of case fatality rate and prevalence of hepatitis B and C virus in people living with HIV in Africa.**
(PDF)

**S1 Fig. Funnel chart for publications of the hepatitis B virus case fatality rate in PLHIV in Africa.**
(PDF)

**S2 Fig. Prevalence estimate of hepatitis B virus infection in PLHIV in Africa.**
(PDF)

**S3 Fig. Funnel chart for publications of hepatitis B virus prevalence in PLHIV in Africa.**
(PDF)

**S4 Fig. Prevalence estimate of hepatitis C virus infection in PLHIV in Africa.**
(PDF)

**S5 Fig. Funnel chart for publications of hepatitis C virus prevalence in PLHIV in Africa.**
(PDF)

**S6 Fig. Funnel chart for publications of hepatitis B/C prevalence in PLHIV in Africa.**
(PDF)

**S1 Text. Reference list of included studies on the prevalence of HBV and/or HCV in PLHIV in Africa.**
(PDF)

## Author Contributions

**Conceptualization:** Raoul Kenfack-Momo, Sebastien Kenmoe, Richard Njouom.

**Data curation:** Raoul Kenfack-Momo, Sebastien Kenmoe, Guy Roussel Takuissu, Jean Thierry Ebogo-Belobo, Cyprien Kengne-Ndé, Donatien Serge Mbaga, Serges Tchatchouang, Martin Gael Oyono, Josiane Kenfack-Zanguim, Robertine Lontuo Fogang, Chris Andre Mbongue Mikangue, Elisabeth Zeuko'o Menkem, Juliette Laure Ndzie Ondigui, Ginette Irma Kame-Ngasse, Jeannette Nina Magoudjou-Pekam, Jean Bosco Taya-Fokou, Arnol Bowo-Ngandji.

**Formal analysis:** Sebastien Kenmoe, Cyprien Kengne-Ndé.

**Funding acquisition:** Sebastien Kenmoe.

**Methodology:** Raoul Kenfack-Momo, Sebastien Kenmoe, Guy Roussel Takuissu, Jean Thierry Ebogo-Belobo, Cyprien Kengne-Ndé, Donatien Serge Mbaga, Serges Tchatchouang, Martin Gael Oyono, Josiane Kenfack-Zanguim, Robertine Lontuo Fogang, Chris Andre Mbongue Mikangue, Elisabeth Zeuko'o Menkem, Juliette Laure Ndzie Ondigui, Ginette Irma Kame-Ngasse, Jeannette Nina Magoudjou-Pekam, Jean Bosco Taya-Fokou, Arnol Bowo-Ngandji, Seraphine Nkie Esemu, Diane Kamdem Thiomo, Paul Moundipa Fewou, Lucy Ndip, Richard Njouom.

**Project administration:** Sebastien Kenmoe.

**Supervision:** Sebastien Kenmoe, Richard Njouom.

**Validation:** Raoul Kenfack-Momo, Sebastien Kenmoe, Guy Roussel Takuissu, Jean Thierry Ebogo-Belobo, Donatien Serge Mbaga, Serges Tchatchouang, Martin Gael Oyono, Josiane Kenfack-Zanguim, Robertine Lontuo Fogang, Chris Andre Mbongue Mikangue, Elisabeth Zeuko'o Menkem, Juliette Laure Ndzie Ondigui, Ginette Irma Kame-Ngasse, Jeannette Nina Magoudjou-Pekam, Jean Bosco Taya-Fokou, Arnol Bowo-Ngandji, Seraphine Nkie Esemu, Diane Kamdem Thiomo, Paul Moundipa Fewou, Lucy Ndip, Richard Njouom.

**Writing – original draft:** Raoul Kenfack-Momo, Sebastien Kenmoe.

**Writing – review & editing:** Raoul Kenfack-Momo, Sebastien Kenmoe, Guy Roussel Takuissu, Jean Thierry Ebogo-Belobo, Cyprien Kengne-Ndé, Donatien Serge Mbaga, Serges Tchatchouang, Martin Gael Oyono, Josiane Kenfack-Zanguim, Robertine Lontuo Fogang, Chris Andre Mbongue Mikangue, Elisabeth Zeuko'o Menkem, Juliette Laure Ndzie Ondigui, Ginette Irma Kame-Ngasse, Jeannette Nina Magoudjou-Pekam, Jean Bosco Taya-Fokou, Arnol Bowo-Ngandji, Seraphine Nkie Esemu, Diane Kamdem Thiomo, Paul Moundipa Fewou, Lucy Ndip, Richard Njouom.

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
