## [Decision Letter · Decision Letter 0]

20 Apr 2022

PONE-D-22-08184Epidemiology of hepatitis B virus and/or hepatitis c virus infections among people living with human immunodeficiency virus in Africa: a systematic review and meta-analysis.PLOS ONE

Dear Dr. Kenmoe,

Thank you for submitting your manuscript to PLOS ONE. After careful consideration, we feel that it has merit but does not fully meet PLOS ONE’s publication criteria as it currently stands. Therefore, we invite you to submit a revised version of the manuscript that addresses the several points raised during the review process.

We look forward to receiving your revised manuscript.

Kind regards,

Isabelle Chemin, PhD

Academic Editor

PLOS ONE

Journal Requirements:

Reviewers' comments:

Reviewer's Responses to Questions

**Comments to the Author**

1. Is the manuscript technically sound, and do the data support the conclusions?

Reviewer #1: Yes

Reviewer #2: Yes

2. Has the statistical analysis been performed appropriately and rigorously? 

Reviewer #1: Yes

Reviewer #2: Yes

3. Have the authors made all data underlying the findings in their manuscript fully available?

Reviewer #1: Yes

Reviewer #2: Yes

4. Is the manuscript presented in an intelligible fashion and written in standard English?

Reviewer #1: Yes

Reviewer #2: Yes

5. Review Comments to the Author

Reviewer #1: Thank you for a very good manuscript. This is very thorough and all expected aspects for the systematic review and metanalysis have been presented. I find very tittle to comment on because the limitations have been identified and discussed too.

So just few thoughts for consideration:

1. the period for the review, dies it end 2021? Or it was 2022 as stated in the manuscript?

2. Why did you limit the language to English and French? These are not the only languages spoken officially in Africa. This limitation is not mentioned at all in the manuscript. Kindly address this.

3. There are few typographical errors which needs to be corrected so kindly go through

4 In the discussion, I expected some recommendations concerning the quality of the publications.

Reviewer #2: Thank you for the opportunity to review this manuscript. It is well performed and the data is well presented. The finding are relevant and important. I have just a few comments to make:

1. The issue of epidemiology, notably HBV-HIV is addressed in the discussion. I may suggest to the authors that it makes somewhat more sense to introduce the issue of HBV being mostly acquired in childhood in sub Saharan Africa, with HIV later in adulthood. This is true for most with HIV-HBV co-infection and may well add to the burden of liver disease. The reasons as to why the HBV rate is somewhat higher in the co-infected population, perhaps warrants some more discussion, although understandably the reasons are not entirely clear.

2. The authors comments on HBV DNA presence in co-infection as a metric [Twelve studies reported the HBV infection prevalence based on the detection of HBV DNA; 17.1% [95% CI=11.4-23.7]. I assume these are ART naive patients in these studies - as most ART would likely contain if not lamivudine, but tenofovir ?

3. Would there be any value looking at HIV-HCV in sub-populations e.g. PWID in the manuscript?

6. PLOS authors have the option to publish the peer review history of their article (what does this mean?). If published, this will include your full peer review and any attached files.

Reviewer #1: No

Reviewer #2: No

---

## [Author Response · Author response to Decision Letter 0]

27 Apr 2022

Review Comments to the Author

Reviewer #1: Thank you for a very good manuscript. This is very thorough and all expected aspects for the systematic review and metanalysis have been presented. I find very tittle to comment on because the limitations have been identified and discussed too. 

Authors: We thank the reviewer for this appreciation.

1. the period for the review, dies it end 2021? Or it was 2022 as stated in the manuscript? 

Authors: As indicated in the manuscript, the study ends in 2022. Indeed, our search strategy was first applied in the databases in February 2021. Due to the delay in finalizing the review we had to reapply our search strategy in January 2022 as indicated in the methodology section. 

2. Why did you limit the language to English and French? These are not the only languages spoken officially in Africa. This limitation is not mentioned at all in the manuscript. Kindly address this.

Authors: We thank the reviewer for this suggestion. We added this limit in the discussion section.

3. There are few typographical errors which needs to be corrected so kindly go through

Authors: Thank you, we have carefully revised it. 

4. In the discussion, I expected some recommendations concerning the quality of the publications.

Authors: Thank you for this suggestion. Half of the studies in our review were at moderate risk of bias. Our sensitivity analysis, that we added in the result section, showed that this risk of bias did not affect our overall results.

“Although about half of the studies had a moderate risk of bias, sensitivity analyzes showed no difference between the overall results and the results including only the studies with a low risk of bias (Table 1).”

Reviewer #2: Thank you for the opportunity to review this manuscript. It is well performed and the data is well presented. The finding are relevant and important.

Authors: Thank you for this appreciation.

1. The issue of epidemiology, notably HBV-HIV is addressed in the discussion. I may suggest to the authors that it makes somewhat more sense to introduce the issue of HBV being mostly acquired in childhood in sub Saharan Africa, with HIV later in adulthood. This is true for most with HIV-HBV co-infection and may well add to the burden of liver disease. The reasons as to why the HBV rate is somewhat higher in the co-infected population, perhaps warrants some more discussion, although understandably the reasons are not entirely clear. 

Authors: Thank you for this comment which we fully agree. We have added the comment below in the discussion to support this aspect. 

“Although this specific transmission pattern in Africa of HBV and HIV in childhood and adulthood respectively is true for most cases, it is evident that many other reasons not fully described and/or clear exist and warrant more attention.”

2. The authors comments on HBV DNA presence in co-infection as a metric [Twelve studies reported the HBV infection prevalence based on the detection of HBV DNA; 17.1% [95% CI=11.4-23.7]. I assume these are ART naive patients in these studies - as most ART would likely contain if not lamivudine, but tenofovir ? 

Authors: Thank you for the comment for which we fully agree. After verification, 4 of the studies focus obviously on ART naive patients, 2 studies have a mixture of naive and on ART subjects, and for the remaining 6 studies the ART status of the patients is unclear and/or not reported.

3. Would there be any value looking at HIV-HCV in sub-populations e.g. PWID in the manuscript? 

Authors: thank you for this suggestion. We added a paragraph in the discussion section about this.

---

## [Editor Report · Decision Letter 1]

18 May 2022

Epidemiology of hepatitis B virus and/or hepatitis C virus infections among people living with human immunodeficiency virus in Africa: a systematic review and meta-analysis.

PONE-D-22-08184R1

Dear Dr. KENMOE,

We’re pleased to inform you that your manuscript has been judged scientifically suitable for publication and will be formally accepted for publication once it meets all outstanding technical requirements.

Kind regards,

Isabelle Chemin, PhD

Academic Editor

PLOS ONE
---

## [Editor Report · Acceptance letter]

20 May 2022

PONE-D-22-08184R1 

Epidemiology of hepatitis B virus and/or hepatitis c virus infections among people living with human immunodeficiency virus in Africa: a systematic review and meta-analysis. 

Dear Dr. Kenmoe:

I'm pleased to inform you that your manuscript has been deemed suitable for publication in PLOS ONE. Congratulations! Your manuscript is now with our production department. 

Kind regards, 

on behalf of

Mrs Isabelle Chemin 

Academic Editor

PLOS ONE